

# Snowpack nitrate photolysis drives the summertime atmospheric nitrous acid (HONO) budget in coastal Antarctica

Amelia M. H. Bond[1,2], Markus M. Frey[1], Jan Kaiser[2], Jörg Kleffmann[3], Anna E. Jones[1], and Freya A. Squires[1]

[1]British Antarctic Survey, Natural Environment Research Council, Cambridge, UK
[2]Centre for Ocean and Atmospheric Sciences, School of Environmental Sciences, University of East Anglia, Norwich, UK
[3]Department of Physical and Theoretical Chemistry, Faculty for Mathematics and Natural Sciences, University of Wuppertal, Germany

**Correspondence:** Amelia Bond (amend37@bas.ac.uk)

**Abstract.** Measurements of atmospheric nitrous acid (HONO) amount fraction and flux density above snow were carried out using a long path absorption photometer at Halley station in coastal Antarctica between 22 January and 3 February 2022. The mean $\pm 1\sigma$ HONO amount fraction was $(2.1 \pm 1.5)\,\mathrm{pmol\,mol^{-1}}$ and showed a diurnal cycle (range $1.0 - 3.2\,\mathrm{pmol\,mol^{-1}}$) with a maximum at solar noon. These HONO amount fractions are generally lower than have been observed at other Antarctic

locations. The flux density of HONO from the snow, measured between 31 January and 1 February 2022, was between 0.5 and $3.4 \times 10^{12}\,\mathrm{m^{-2}\,s^{-1}}$, and showed a decrease during the night. The measured flux density is at the upper limit of the calculated HONO production rate from photolysis of nitrate present in the snow. A simple box model of HONO sources and sinks showed that the flux of HONO from the snow makes a $> 10$ times larger contribution to the HONO budget than its formation through the reaction of OH and NO. Ratios of these HONO amount fractions to $NO_x$ measurements made in summer 2005 are

low $(0.15 - 0.35)$, which we take as an indication of our measurements being comparatively free from interferences. Further calculations suggest that HONO photolysis could produce up to $12\,\mathrm{pmol\,mol^{-1}\,h^{-1}}$ of OH, approximately half that produced by ozone photolysis, which highlights the importance of HONO snow emissions as an OH source in the atmospheric boundary layer above Antarctic snowpacks.

## 1 Introduction

Photolysis of nitrous acid (HONO) is a crucial polar boundary layer source of the hydroxyl radical (OH), a daytime oxidant that is important for the removal of many pollutants, including the greenhouse gas methane ($CH_4$) (Kleffmann, 2007).

$$HONO + h\nu \rightarrow OH + NO \tag{R1}$$

On a global scale, OH radical formation is usually controlled by ozone ($O_3$) photolysis followed by reaction with water vapour:

$$O_3 + h\nu \rightarrow O_2 + O(^1D) \tag{R2}$$



$$O(^1D) + H_2O \rightarrow 2\,OH. \tag{R3}$$

The OH production by $O_3$ photolysis is expected to be limited in the polar regions because in a cold atmosphere the water vapour concentration is low (Davis et al., 2008). It has been established that sunlit polar snow-packs are an important source

of OH precursors for the lower atmosphere including $NO_x$ (Honrath et al., 1999; Jones et al., 2000) and HONO (Zhou et al., 2001), as well as formaldehyde ($CH_2O$) and hydrogen peroxide ($H_2O_2$) (Hutterli et al., 2002, 2004; Frey et al., 2005). Unexpectedly high HONO amount fractions have been measured above snow surfaces in polar regions (Zhou et al., 2001; Honrath et al., 2002; Beine et al., 2001, 2002; Dibb et al., 2002, 2004; Kerbrat et al., 2012; Legrand et al., 2014) and also at mid-latitudes (Kleffmann et al., 2002; Kleffmann and Wiesen, 2008; Michoud et al., 2015; Chen et al., 2019).

30  In the boundary layer HONO is formed through the homogeneous reaction of OH and NO:

$$OH + NO \rightarrow HONO. \tag{R4}$$

At Arctic and Antarctic Plateau locations this has been found to have a lower contribution to the HONO budget than emission from the snow (Villena et al., 2011; Legrand et al., 2014). However, the importance of different HONO sources is less clear in coastal Antarctica (Beine et al., 2006). The dominant HONO loss process is photolysis (R1), but it is also lost through reaction

with OH:

$$HONO + OH \rightarrow H_2O + NO_2. \tag{R5}$$

 The exact mechanism for HONO release from snow is not understood and models of HONO sources and sinks often cannot rationalise the measured HONO amount fractions (Villena et al., 2011; Legrand et al., 2014). Nitrate photolysis in snow produces nitrite ($NO_2^-$):

$$NO_3^- + h\nu \rightarrow NO_2^- + O \tag{R6}$$

which can be protonated to form HONO (Honrath et al., 2000; Zhou et al., 2001):

$$NO_2^- + H^+ \rightarrow HONO. \tag{R7}$$

Correlations have been observed between snow nitrate concentrations and HONO formation (Dibb et al., 2002; Legrand et al., 2014). Several studies also report reduced HONO production from alkaline snow, which supports this mechanism (Beine et al.,

2005, 2006; Amoroso et al., 2006). However, the dominant product from nitrate photolysis is nitrogen dioxide ($NO_2$):

$$NO_3^- + h\nu \rightarrow NO_2 + O^- \tag{R8}$$

which can undergo hydrolysis to produce HONO via disproportionation (Finlayson-Pitts et al., 2003):

$$2\,NO_2 + H_2O \rightarrow HONO + HNO_3 \tag{R9}$$



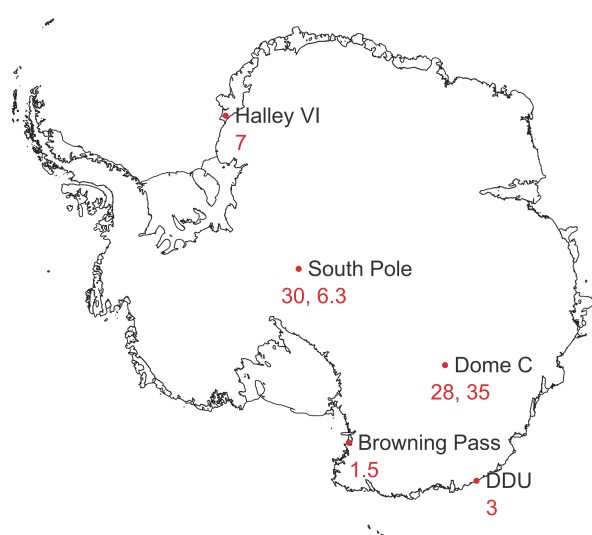

**Figure 1.** A map showing the mean atmospheric HONO amount fractions (in $pmol\,mol^{-1}$) measured previously in the Antarctic lower troposphere during summer (Dibb et al., 2004; Beine et al., 2006; Liao et al., 2006; Bloss et al., 2010; Kerbrat et al., 2012; Legrand et al., 2014).

or reactions on organic surfaces in the snow (Ammann et al., 2005):

$$NO_2 + organics \rightarrow HONO + products. \tag{R10}$$

Such reactions are accelerated by sunlight (George et al., 2005):

$$NO_2 + h\nu + organics \rightarrow HONO + products. \tag{R11}$$

The reaction of $NO_2$ on photosensitised organics (R11) has been found to occur much faster than the disproportionation reaction (R9) (Stemmler et al., 2006). HONO formation from humic acid-doped ice films under a flow of $NO_2$ was found to scale with both the $NO_2$ and humic acid concentration (Beine et al., 2008; Bartels-Rausch et al., 2010). During their measurement campaign in Alaska, Villena et al. (2011) found a correlation between their calculated HONO snow-source strength and $[NO_2] \times J(NO_2)$, but not $[NO_3^-] \times J(O(^1D))$, suggesting that conversion of $NO_2$ on photosensitised organic surfaces in the snow is the likely source of HONO (R11).

HONO amount fractions have been measured at both Arctic and Antarctic locations, as well as above mid-latitude snow covered areas. In Antarctica, HONO has been detected at inland and coastal locations, summarised in Fig. 1. Previous results from Halley Research Station, a coastal, ice-shelf location, gave average HONO amount fractions of $7\,pmol\,mol^{-1}$ during the CHABLIS campaign in January - February 2005 (Bloss et al., 2010), but this was thought to be an overestimate due





to chemical interferences in the wet-chemical HONO instrument used (Jones et al., 2011). On the Antarctic Plateau HONO amount fractions are higher. At the South Pole, up to 18 $pmol\,mol^{-1}$ HONO was measured by laser induced fluorescence (LIF)

(Liao et al., 2006) and at Dome Concordia (Dome C), more recent measurements using a LOng Path Absorption Photometer yielded HONO amount fractions of 28 $pmol\,mol^{-1}$ in summer 2010/11 (Kerbrat et al., 2012) and 35 $pmol\,mol^{-1}$ in 2011/12 (Legrand et al., 2014). A strong diurnal cycle of HONO was observed in both measurement periods, with enhancements in the morning and evening suggesting a photochemical source. In contrast, at Dumont D'Urville (DDU), a coastal site without snow cover, HONO amount fractions were much lower, with a mean of 3 $pmol\,mol^{-1}$ and no diurnal variation. However, the arrival

of inland Antarctic air masses at DDU coincided with higher HONO amount fractions supporting the existence of a HONO source in the continental snowpack (Kerbrat et al., 2012).

There have been significant issues with the overestimation of atmospheric HONO amount fractions by various measurement techniques due to interferences. Measurements made at the South Pole with mist chamber sampling followed by ion chromatography analysis (MC/IC) gave 6 times higher values than those made by LIF (Dibb et al., 2004; Liao et al., 2006). At

Halley, the wet chemical method (scrubbing HONO into water, azo dye derivatisation, followed by optical detection) did not allow for interference removal and hence the HONO amount fractions were overestimated (Jones et al., 2011). In contrast, the two-channel concept of the LOng Path Absorption Photometer (LOPAP) used at Dome C and DDU is expected to correct for most interferences. In addition, the external sampling unit of this instrument minimises sampling artefacts, for example, those in sampling lines typically used for other HONO instruments. However, the high HONO amount fractions observed at Dome

C were partially explained with potential interference of peroxynitric acid ($HNO_4$). The interference of $HNO_4$ in the LOPAP instrument has not been systematically studied, and the documented $HNO_4$ interference of around 15 % may become an issue at lower temperatures due to its longer lifetime with respect to thermal decomposition (Legrand et al., 2014).

Further investigation is clearly needed to better understand HONO sources and sinks in the polar boundary layer, and the implications for the $HO_x$ budget. This paper presents measurements of HONO amount fractions and flux densities made at

Halley during austral summer 2021/22. A LOPAP instrument was used for this study to minimise interferences and sampling artefacts. The results are rationalised using knowledge of possible HONO sources, and the potential of HONO as an OH source to the boundary layer at Halley will be discussed.

## 2 Site and methods

### 2.1 Site

Our measurement campaign took place between 22 January and 3 February 2022 at Halley VI Research Station (75°34'5" S, 25°30'30" W), which is located on the the Brunt Ice Shelf, Antarctica at 32 m above mean sea level (Fig. 2). This work was carried out in the Clean Air Sector (CAS), 1.5 km south of the main station buildings, avoiding the influence of pollution from station generators and vehicles. The instrument to detect atmospheric HONO was housed in a container at ground level, 10 m north of the CAS laboratory. The average wind speed was $10\,m\,s^{-1}$, and reached up to a maximum of $26\,m\,s^{-1}$. The dominant



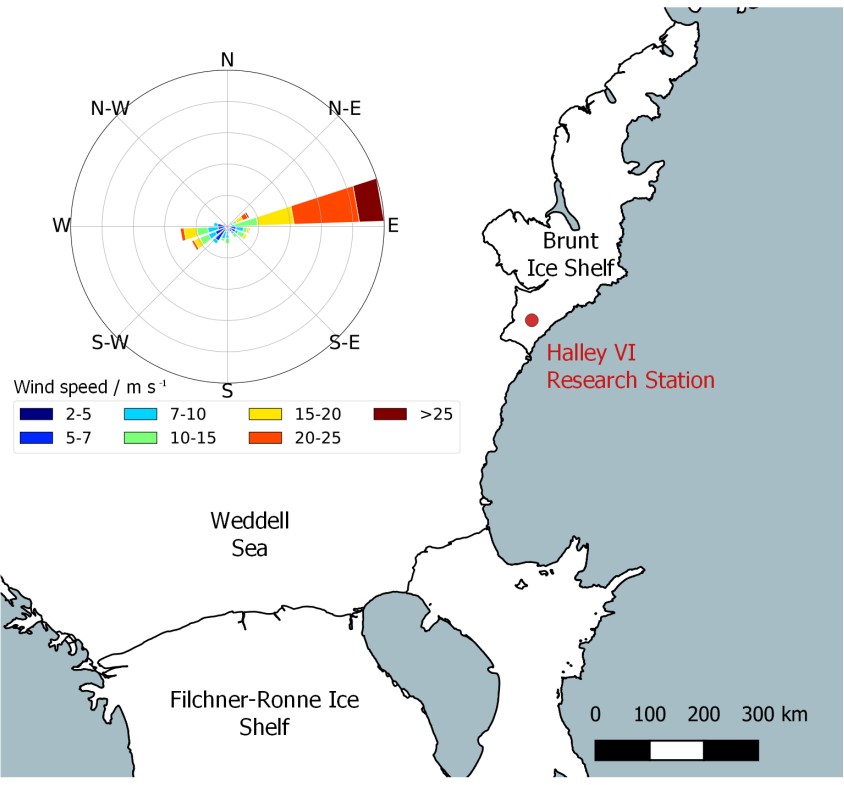

**Figure 2.** A map showing the location of Halley on the Brunt Ice Shelf, Antarctica, as well as a wind-rose plot for the period of the measurement campaign.

wind direction during the campaign was east, see Fig. 2, and the air temperature was between $-13$ and $+1\,°C$, with a mean of $-4\,°C$. All times are given in UTC, where local noon and midnight were at 1400 and 0200, respectively.

## 2.2   Methods

HONO was detected using a LOng Path Absorption Photometer (LOPAP, QUMA Elektronik & Analytik GmbH) which has been described in detail elsewhere (Heland et al., 2001; Kleffmann et al., 2002). Briefly, the instrument works by first collecting
HONO in a stripping coil, housed in a temperature-controlled external sampling unit, by a fast chemical reaction in an acidic (pH = 0) sulfanilamide solution (reagent 1, $1\,g\,L^{-1}$, lower than originally proposed, see von der Heyden et al. (2022)). HONO is initially converted into $NO^+$ which forms a diazonium salt by reaction with sulfanilamide. Due to the fast chemical reaction, much shorter gas-liquid contact times are applied (4-ring coil) compared to other wet-chemical HONO instruments (typically $\geq$ 10-ring coils), which require physical solubility equilibrium. This approach minimises sampling of interferences. In addition,
the acidic sampling conditions slow down most known interfering reactions, which are faster under the neutral to alkaline



conditions typically used in other wet-chemical instruments (Kleffmann and Wiesen, 2008). The solution is then pumped via a 3 m-long temperature-controlled reagent line to the main instrument, in which an azo-dye is formed by reaction with a $0.1 \, \mathrm{g \, L^{-1}}$ N-(1-naphthyl)-ethylenediamine dihydrochloride (NED) solution (reagent 2). The dye is detected in long path absorption tubing (path length 5 m) by a spectrometer (Ocean Optics SD2000) at 550 nm. The dye concentration can be related

to the atmospheric HONO amount fraction by carrying out calibrations with nitrite solutions of known concentrations and knowing the sample air to liquid flow rate ratio.

The sampling unit is made up of two stripping coils in series such that HONO and some interferences are taken up in the first coil, followed by only interferences in the second. The interferences should be taken up to the same extent in both channels so that the HONO amount fraction can be calculated by subtracting the channel 2 signal from channel 1 (Heland

et al., 2001). The instrument has been studied for the effect of various possible interfering species, including NO, $NO_2$, $O_3$, peroxyacetyl nitrate (PAN), $HNO_3$ and even more complex mixtures of volatile organic compounds (VOCs) and $NO_x$ in diesel engine exhaust fumes (Heland et al., 2001; Kleffmann et al., 2002). The instrument gave good agreement with the differential optical absorption spectroscopy (DOAS) technique under complex urban and smog chamber conditions (Kleffmann et al., 2006). However, comparison of the instrument under pristine polar conditions is still an open issue. In the present study, the

average interference was 40 % of the channel 1 signal, highlighting the importance of using a two-channel instrument, in excellent agreement with other studies of LOPAP instruments under polar conditions and at high mountain sites (Kleffmann and Wiesen, 2008; Villena et al., 2011).

During the campaign the LOPAP was calibrated every 5 days using nitrite solutions of known concentration ($2 \times 10^{-3}$ and $8 \times 10^{-4} \, \mathrm{mg \, L^{-1}}$). To maximise the instrument sensitivity, the gas-to-liquid flow rate ratio was optimised: the gas flow rate

was set to $2 \, \mathrm{L \, min^{-1}}$ (298 K, 1 atm) with the internal mass flow controller and checked frequently using a flow meter (DryCal DC-Lite), and the liquid flow rate through the stripping coil for each channel was regularly measured volumetrically and was between 0.15 and $0.18 \, \mathrm{mL \, min^{-1}}$ during the measurement period. Baseline measurements were made every 6 hours using a flow of pure nitrogen (99.998 %, BOC) at the instrument inlet. The detection limit ($3 \, \sigma_{\mathrm{blank}}$) was $0.26 \, \mathrm{pmol \, mol^{-1}}$ for the measurement period.

The LOPAP sampling unit required adaptation for use in cold polar environments. The sampling unit box and the reagent lines between this and the main instrument have been coated in Armaflex insulation; no HONO emission from such insulation materials has been detected (Kerbrat et al., 2012). The stripping coil and tubing to the sampling unit are temperature-controlled by a flow from a water bath (Thermo Haake K10 with DC10 circulator). The temperature of the sampling unit was kept at $+18 \, ^\circ \mathrm{C}$.

The instrument's external sampling unit was mounted 0.4 m above the snow, 2.4 m from the container entrance, and pointed into the dominant wind direction (east). For most of the campaign the sampling unit was stationary, except for a period of 12 hours between 1500 on 31 January and 0300 on 1 February 2022 when the height was changed in regular intervals between 0.24 and 1.26 m above the snow in order to measure the HONO gradients needed to estimate vertical fluxes. An automatic elevator, built in-house, was used to raise and lower the sampling unit every 15 minutes (travel time = 1 minute), meaning there





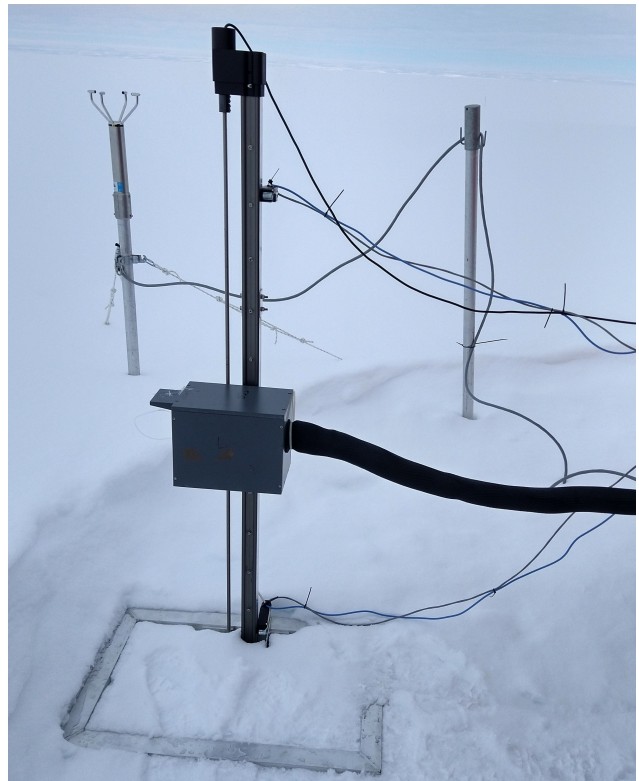

**Figure 3.** An image of the elevator used to raise and lower the LOPAP sampling unit in order to estimate the air-snow HONO flux density using the flux-gradient method.

was no human involvement in moving the sampling unit and no tubing was needed to sample at different heights. Such tubing provides an artificial surface for HONO formation (Villena et al., 2011). The elevator is depicted in Fig. 3.

## 2.3 Ancillary measurements

The surface ozone amount fraction was measured simultaneously from the CAS lab by UV absorption (Thermo Scientific Model 49i Ozone Analyzer). Data were collected at a 10 s interval, quality controlled and then averaged to 1-minute for this

analysis. Instrument limit of detection (LOD) was taken to be $3\sigma$ of 2 hours of 10 s measurements of zero air. This was calculated to be 0.38 nmol mol$^{-1}$. The analyser inlet pointed east and was located at 8 m above the snow surface.

For the HONO flux density calculation the wind speed and direction was measured with a 2D sonic anemometer (Gill Wind Observer 70) located 1.5 m south of the sampling unit and 1 m above the snow. The temperature gradient was measured with two thermometers (TME Ethernet Thermometer) mounted on the vertical post of the elevator at 0.05 and 1.26 m above the

snow surface. During the campaign the incoming shortwave solar radiation $(300 - 2800\,\text{nm})$ was measured by a net radiometer





(Kipp & Zonen, CNR4) located at the main station (1.5 km from the HONO sampling site). The ozone column density was measured with a Dobson spectrophotometer also at the main station.

## 2.4 Flux Calculations

The flux-gradient method was used to determine the HONO flux density following a similar approach as done previously

for $NO_x$ in Antarctica (Jones et al., 2001; Frey et al., 2013). By measuring the HONO amount fraction at two heights, the concentration gradient can be found and is related to the flux density by

$$F = -K_c \frac{dc}{dz} \tag{1}$$

for which $K_c$ is the turbulent diffusion coefficient (in $m^2\,s^{-1}$) of a chemical tracer. In the atmospheric boundary layer, $K_c$ may be approximated by the eddy diffusion coefficient for heat, $K_h$ (Jacobson, 2005). It should be noted that a negative gradient in

amount fraction will result in a positive flux density, equivalent to emission from the snow.

Monin-Obukhov Similarity Theory (MOST) is used to parameterise fluxes in the surface layer, about 10 % of the depth of the atmospheric boundary layer (Stull, 1988), where turbulent fluxes are assumed to be independent of height. The flux density can be calculated by:

$$F = \frac{\kappa u_* [c(z_1) - c(z_2)]}{\int_{z_1}^{z_2} \Phi_h\left(\frac{z}{L}\right)\frac{dz}{z}} \tag{2}$$

where $\kappa$ is the von Karman constant (set to 0.4), $u_*$ is the friction wind velocity, found from wind speed measurements, and $c(z)$ is the HONO amount fraction at height $z$. $\int_{z_1}^{z_2} \Phi_h\left(\frac{z}{L}\right)\frac{dz}{z}$ is the integrated stability function for heat, a function of $\frac{z}{L}$ where $L$ is the Obukhov length. The full derivation of Eq. (2) is in appendix A.

The application of MOST requires certain conditions to be met (Frey et al., 2013): (a) the flux density is constant between the two measurement heights, (b) the lower inlet height is above the surface roughness length, (c) the upper measurement height

is within the surface layer and (d) the measurement heights is far enough apart for the detection of a significant difference in amount fraction.

For (a) the chemical lifetime ($\tau_{chem}$) with respect to photolytic loss was compared to the transport time ($\tau_{trans}$) between the two measurement heights. If $\tau_{chem}$ is much larger than $\tau_{trans}$, then the flux density can be assumed to be constant. $\tau_{chem}$, found from the inverse of the photolysis rate coefficient, $J(HONO)$, was between 10 and 80 minutes. The transport time can be

estimated by (Jacobson, 2005):

$$\tau_{trans} = (z_2 - z_1) \int_{z_1}^{z_2} \frac{dz}{K_h} = (z_2 - z_1) \frac{\int_{z_1}^{z_2} \Phi_h\left(\frac{z}{L}\right)\frac{dz}{z}}{\kappa u_*}. \tag{3}$$

The transport time between 0.24 and 1.26 m above the snow for the flux measurement period at Halley was between 16 and 29 seconds. In all cases the lifetime was significantly longer than the transport time, meaning the flux density can be assumed to be constant between the two heights.





The lower measurement height was 0.24 m, which is significantly above the surface roughness length of $(5.6 \pm 0.6) \times 10^{-5}$ m measured previously at Halley (King and Anderson, 1994).

During the Antarctic summer the boundary layer height at Halley is regularly stable making it difficult to define (Anderson and Neff, 2008). Previous analysis of sodar (sound detection and ranging) measurements has suggested that the boundary layer at Halley in summer is consistently above 40 m (Jones et al., 2008). The equations of both Pollard et al. (1973) and Zilitinkevich
and Baklanov (2002) have been used to estimate the mixing height at Antarctic locations (South Pole (Neff et al., 2008), Dome C (Frey et al., 2013)). Though they are unlikely to predict the height accurately, these equations can provide a useful estimate of the minimum boundary layer height. For the period in question, this is calculated to be 75 and 95 m (Pollard et al. (1973) and Zilitinkevich and Baklanov (2002), respectively). When temperature profiles recorded by daily weather balloon launches during the measurement campaign show a temperature inversion (Stull, 1988), this was above 100 m. Therefore, the upper
measurement height, 1.26 m, was very likely within the surface layer.

A $t$-test confirmed that the amount fraction difference between the two heights ($\Delta y$) was significant ($p < 0.01$).

All of the above criteria were satisfied for the measurement period, so MOST was used to calculate the flux density by the method described above.

## 2.5   Photolysis rates

The rate coefficient of photochemical reactions can be calculated from

$$J = \int_{\lambda_1}^{\lambda_2} \sigma(\lambda, T) \varphi(\lambda, T) F(\lambda) \, \mathrm{d}\lambda \tag{4}$$

where $\sigma$ and $\varphi$ are the absorption cross-section and quantum yield for the photolysis reaction of interest, functions of wavelength ($\lambda$) and temperature ($T$). $F$ is the actinic flux derived from the TUV radiation model over the wavelength range 300 to 1200 nm using measured ozone column density and assuming clear sky conditions (Madronich and Flocke, 1999; Lee-Taylor
and Madronich, 2002). The calculated $J$ values were then scaled by the ratio of measured and modelled incoming shortwave solar radiation to account for non-clear sky conditions (see Fig. 5). It is noted that the wavelength ranges of modelled and measured radiation are not exactly the same, but the contribution of wavelengths $> 1200$ nm is expected to be small.

## 3   Results

### 3.1   HONO amount fraction

HONO amount fractions measured at Halley were between $< 0.3$ and 14 pmol mol$^{-1}$ (Fig. 4), with a mean of 2.1 pmol mol$^{-1}$.

These HONO amount fractions are some of the lowest ever observed in Antarctica (see Table 1) and are only the second series of HONO observations at an Antarctic coastal ice-shelf location. When measurements were attempted once before at Halley, it was thought that the HONO amount fractions were overestimated (Clemitshaw, 2006; Jones et al., 2011). The HONO data collected in this study support this suggestion as the mean is 2.1 pmol mol$^{-1}$ compared with around 7 pmol mol$^{-1}$



| Location | Mean $y$(HONO)/ pmol mol$^{-1}$ | Range $y$(HONO)/ pmol mol$^{-1}$ | Measurement technique | Campaign dates | Reference |
|---|---|---|---|---|---|
| Halley | 2.1 | $< 0.3 - 14.0$ | LOPAP | Jan - Feb 2022 | This work |
| Halley | 7 | $-$ | Scrubbing HONO into water azo dye derivatisation and detection | Jan - Feb 2005 | Bloss et al. (2010) |
| DDU | 3 | $0 - 14$ | LOPAP | Feb 2011 | Kerbrat et al. (2012) |
| Browning Pass | 1 to 2 | $0 - 7$ | Phosphate buffer sampling, azo dye derivatisation | Nov 2004 | Beine et al. (2006) |
| South Pole | 30 | $5 - 71$ | Mist chamber sampling, ion chromatography analysis | Dec 2000 | Dibb et al. (2004) |
| South Pole | 6.3 | $< 3 - 18.2$ | LIF | Nov - Dec 2003 | Liao et al. (2006) |
| Dome C | 30.4 35 | $5 - 59$ | LOPAP | Dec 2010 - Jan 2011 Dec 2011 - Jan 2012 | Legrand et al. (2014) |

**Table 1.** Previous summertime measurements of atmospheric HONO amount fractions ($y$) in Antarctica.

measured in January-February 2005 (Bloss et al., 2010). During this measurement period the average interference was 40 % of the channel 1 value, and occasionally $> 100\%$, showing that HONO would be significantly overestimated without the two-channel sampling unit of the LOPAP.

At other coastal sites the HONO amount fractions are close to those seen in this study. At Browning Pass (Fig. 1), HONO amount fractions were $< 5 \, \text{pmol mol}^{-1}$, though the site conditions are unlike Halley since the snow composition and pH may be affected by rock outcrops nearby (Beine et al., 2006). Small amount fractions were also observed at Dumont D'Urville (DDU), but this was attributed to the fact the site had no snow cover (Kerbrat et al., 2012). Higher amount fractions were observed in continental air masses, likely due to emissions from the snow-pack on the continent.

HONO amount fractions at inland Antarctic locations are predominantly higher than those seen at Halley. At the South Pole, mean HONO amount fractions of $6.3 \, \text{pmol mol}^{-1}$ were measured by LIF (Liao et al., 2006). Dome C HONO amount fractions, measured using a LOPAP, were found to be higher than in most other studies (mean ca. $30 \, \text{pmol mol}^{-1}$) (Legrand et al., 2014). The higher HONO and $NO_x$ amount fractions can be explained by specific conditions on the high Plateau during summer, which include 24-hour sunlight, a shallow and frequently stable boundary layer and very low temperatures (King et al., 2006) leading to low primary production rates for $HO_x$ radicals (Davis et al., 2008). This causes a non-linear $HO_x - NO_x$ chemical



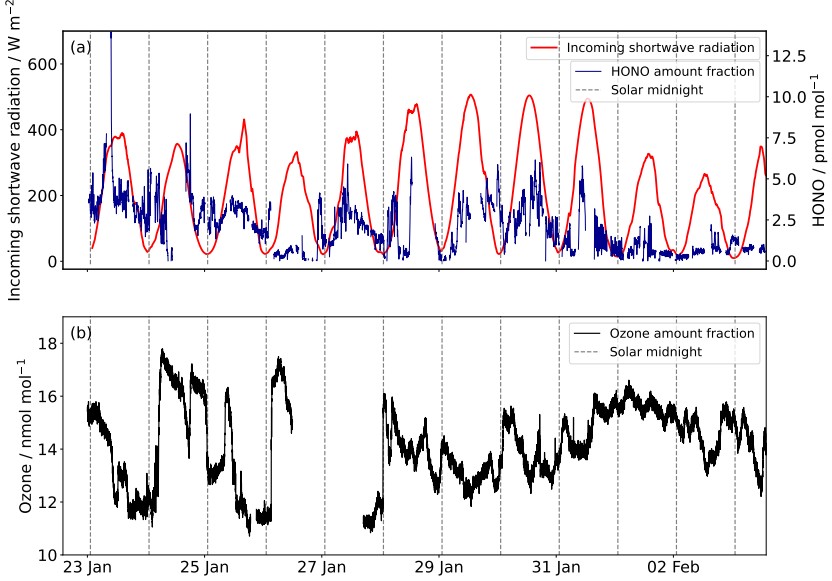

**Figure 4.** (a) The HONO amount fraction at 1-minute resolution recorded at Halley (blue) between 22 January and 3 February 2022 and incoming shortwave radiation between 300 and 2800 nm (red). The dashed lines are solar midnight (0200 UTC). (b) The surface ozone amount fraction.

regime where the $NO_x$ lifetime increases with increasing $NO_x$ as proposed previously (Davis et al., 2008; Neff et al., 2018). Together these factors support increased air-snow recycling and the accumulation of $NO_y$ in the regional boundary layer.

An interference of $HNO_4$ in the LOPAP has been suggested (Kerbrat et al., 2012; Legrand et al., 2014). The LOPAP's response to $HNO_4$ has been investigated in both the laboratory with an $HNO_4$ source and in the field at Dome C by placing a heated tube at the instrument inlet to decompose $HNO_4$. Both showed that the LOPAP partially measures $HNO_4$ as HONO but further investigation is needed to systematically quantify this effect (Legrand et al., 2014). In any case, Dome C is expected
to have a much higher $HNO_4$ amount fraction than Halley due to the fact its lifetime is controlled by thermal decomposition:

$$HNO_4 + M \rightarrow HO_2 + NO_2 + M. \tag{R12}$$

The rate coefficients for thermal decomposition of $HNO_4$ are in Table 2. An average $HNO_4$ lifetime with respect to thermal decomposition of 12 min was calculated from $\frac{1}{k_{12}}$ at Halley. This can be compared to an average lifetime of 21 hours during January 2012 at Dome C (mean temperature $-31\,°C$).
The median diurnal cycle of HONO amount fraction shows a maximum at solar noon (Fig. 5), with a peak-to-peak amplitude of $2\,pmol\,mol^{-1}$. Previous observations of HONO at Halley also showed a diurnal cycle but with a larger day-to-night variation (Clemitshaw, 2006), which is likely an over-estimate of the true variation in HONO amount fractions. The diurnal cycle





**Table 2.** Rate coefficients used in calculations.

| $k$ | Values | Reaction | Ref. |
|---|---|---|---|
| $k_4$ | $k_0 = 7.4 \times 10^{-31} \left(\frac{T}{300\,K}\right)^{-2.4}$ [M] cm$^6$ s$^{-1}$ <br> $k_\infty = 3.3 \times 10^{-11} \left(\frac{T}{300\,K}\right)^{-0.3}$ cm$^3$ s$^{-1}$ <br> $F_c = 0.81$ | R4 | |
| $k_5$ | $2.5 \times 10^{-12}\, e^{\left(\frac{260\,K}{T}\right)}$ cm$^3$ s$^{-1}$ | R5 | Atkinson et al. (2004) <br> IUPAC (last accessed: 2022-12-12) |
| $k_{12}$ | $k_0 = 4.1 \times 10^{-5} e^{\left(\frac{-10650\,K}{T}\right)}$ [M] cm$^3$ s$^{-1}$ <br> $k_\infty = 6.0 \times 10^{15} e^{\left(\frac{-11170\,K}{T}\right)}$ s$^{-1}$ <br> $F_c = 0.4$ | R12 | |
| $k_{10}$ | Gradient of increase in HONO/NO$_x$ at night: $\frac{\Delta \frac{[HONO]}{[NO_x]}}{\Delta t}$ | R10 | Kleffmann et al. (2003) |

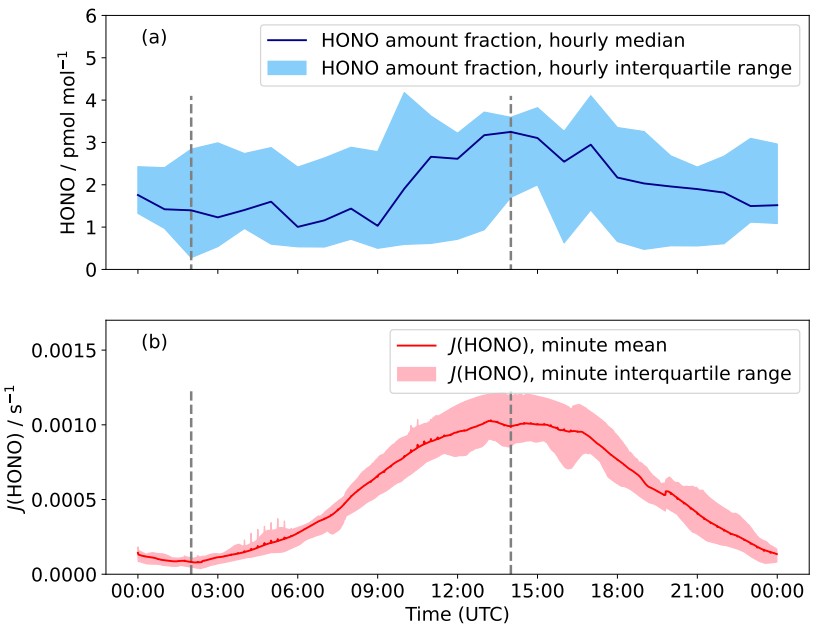

**Figure 5.** (a) The hourly median diurnal cycle in HONO amount fraction for 22 January to 3 February 2022 (blue). The shaded region is the hourly interquartile range and the grey dashed lines solar midnight and noon (0200 and 1400 UTC respectively). (b) $J(HONO)$ (red) which was calculated by the TUV radiation model and then scaled to incoming solar radiation, again the interquartile range is the shaded region.

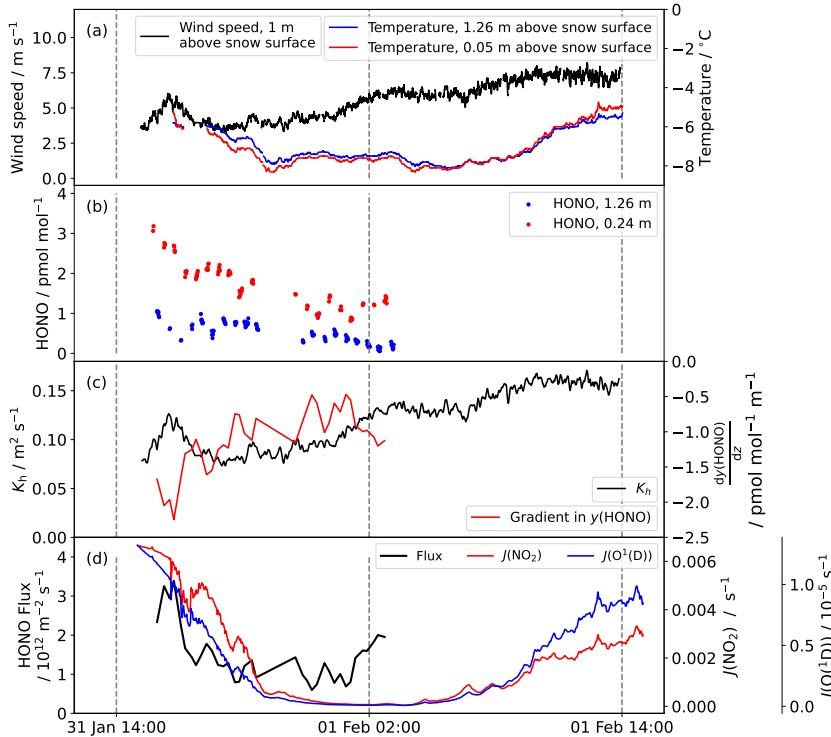

**Figure 6.** (a) The wind speed and temperature recorded at Halley between 31 January and 1 February 2022. (b) The HONO amount fraction measured at 1.26 and 0.24 m above the snow surface. (c) The turbulent diffusion coefficient $K_h$ and the amount fraction gradient calculated from the HONO amount fraction measurements at two heights (note that a negative gradient corresponds to emission of HONO from the snow). (d) The flux density calculated by combining $K_h$ and the amount fraction gradient. The diurnal cycles in $J(\mathrm{NO_2})$ and $J(\mathrm{O}(^1\mathrm{D}))$ calculated by the TUV radiation model, scaled to incident radiation, are also plotted.

observed at Browning Pass compares well with that observed here; the maximum variation was $1\,\mathrm{pmol\,mol^{-1}}$ (Beine et al., 2006). The diurnal cycle of HONO amount fraction at Dome C showed a double peak due to the presence of a strong diurnal

cycle in the boundary layer height; the amount fraction dipped at mid-day when the boundary layer height showed a strong increase (Legrand et al., 2014). The boundary layer height at Halley does not show such a diurnal pattern (King et al., 2006), so the variation is predominantly caused by the photochemical production of HONO peaking at solar noon.





## 3.2 HONO flux density

The measured flux density is plotted in Fig. 6 with the data needed for its calculation: the wind-speed and temperature used to
find the turbulent diffusion coefficient for heat ($K_h$) and the measured amount fraction gradient. The HONO amount fraction
gradient is steep; the amount fraction decreases by about half between 0.24 and 1.26 m above the snow. The flux density varies
between 0.5 and $3.4 \times 10^{12}\,\mathrm{m}^{-2}\,\mathrm{s}^{-1}$ from the snow surface, mainly driven by the amount fraction gradient, and appears to
decrease between solar noon at 1400 UTC and solar midnight at 0200 UTC, suggesting a photochemical snow-pack source.
In the Antarctic, HONO fluxes have previously only been measured at Browning Pass, where larger values were found (mean
upwards flux density of $4.8 \times 10^{12}\,\mathrm{m}^{-2}\,\mathrm{s}^{-1}$). However, occasionally the flux was downwards, equivalent to deposition. This is
likely caused by the atypical snow composition: the snow is only weakly acidic and occasionally alkaline (Beine et al., 2006).
More HONO flux measurements are available from the Arctic; flux densities up to $10^{14}\,\mathrm{m}^{-2}\,\mathrm{s}^{-1}$ from the snow have been
observed (Zhou et al., 2001; Amoroso et al., 2010). However, these Arctic sites are more polluted and therefore there are more
HONO precursors in the snow ($NO_2$, nitrate, organics). Legrand et al. (2014) used measurements of the $NO_x$ flux density at
Dome C and the HONO to $NO_x$ production rate ratio measured in a snow photolysis experiment in the laboratory to estimate
a HONO emission flux density between 5 and $8 \times 10^{12}\,\mathrm{m}^{-2}\,\mathrm{s}^{-1}$, larger than that observed here, likely due to the higher snow
nitrate concentrations at Dome C.

## 4 Discussion

### 4.1 HONO formation mechanisms

HONO formation in the snow-pack, driving the flux to the boundary layer above, is typically attributed to nitrate photolysis.
A HONO flux density from this reaction can be compared to that measured in this study to determine the source of HONO.
The production rate (per area) above snow of reactive nitrogen from snow nitrate photolysis, $P(NO_y)$, can be estimated using
the following equation:

$$P(\mathrm{NO}_y) = \int_0^\infty J(\mathrm{NO_3}^-)[\mathrm{NO_3}^-]\mathrm{d}z \qquad (5)$$

where $J(\mathrm{NO_3}^-)$ is the nitrate photolysis rate coefficient, a function of depth in the snow pack, $z$. This can be approximated
by $J_0(\mathrm{NO_3}^-)\mathrm{e}^{-\frac{z}{z_e}}$ (Chan et al., 2015), where $J_0(\mathrm{NO_3}^-)$ is the photolysis rate coefficient at the snow surface. The e-folding
depth ($z_e$) is between 3.7 and 10 cm (7 cm is used here (Jones et al., 2011)). $[\mathrm{NO_3}^-]$ is the nitrate number concentration (in
units of cm$^{-3}$). To derive HONO production from nitrate photolysis, a HONO yield coefficient $Y(\mathrm{HONO})$ is included (Chen
et al., 2019), and we also explicitly show the conversion from nitrate mass fraction to number concentration:

$$P(\mathrm{HONO}) = \int_0^\infty J_0(\mathrm{NO_3}^-)\mathrm{e}^{-\frac{z}{z_e}}\frac{w(\mathrm{NO_3}^-)\rho_{\mathrm{snow}}N_A}{M(\mathrm{NO_3}^-)}Y(\mathrm{HONO})\mathrm{d}z. \qquad (6)$$

$w(\mathrm{NO_3}^-)$ is the snow nitrate mass concentration (mass per mass of snow), $\rho_{\mathrm{snow}}$ is the snow density, taken as $0.3\,\mathrm{g\,cm^{-3}}$ (Dominé et al., 2008), $N_{\mathrm{A}}$ is Avogadro's number and $M(\mathrm{NO_3}^-)$ is the nitrate molar mass. $J_0(\mathrm{NO_3}^-)$ was found from Eq. (4) using $\mathrm{NO_3}^-$ absorption cross-sections and quantum yields for ice (Chu and Anastasio, 2003), and is $1.2 \times 10^{-7}\,\mathrm{s^{-1}}$ at noon for the measurement period (assuming clear sky conditions).

Daily nitrate mass concentrations in surface snow were measured during the CHABLIS campaign; the mean value for January to February is $(47.1 \pm 17\,\mu\mathrm{g\,L^{-1}})$ (Jones et al., 2011). A yield of $100\,\%$ gives a HONO production rate of $3.7 \times 10^{12}\,\mathrm{m^{-2}\,s^{-1}}$ at a light intensity corresponding to local noon, just above the maximum measured flux density of $3.4 \times 10^{12}\,\mathrm{m^{-2}\,s^{-1}}$, but the yield is unlikely to be as high as $100\,\%$.

There are two product channels for nitrate photolysis (reactions R6 and R8); if it is assumed that all nitrite produced in R6
is converted to HONO, $Y(\mathrm{HONO})$ would be $10\,\%$ because R8 dominates over R6 by a factor of 9 (Chu and Anastasio, 2003). This gives a noon HONO production rate of only $(0.37 \pm 0.30) \times 10^{12}\,\mathrm{m^{-2}\,s^{-1}}$. This is lower than the measured flux density, see Fig. 7. However, it has been suggested that the rate of nitrite production from nitrate photolysis could be as high as that of $\mathrm{NO_2}$ (Benedict and Anastasio, 2017; Benedict et al., 2017) implying $Y(\mathrm{HONO})$ is greater than $10\,\%$. Furthermore, the HONO yield from nitrate photolysis may also include a contribution from $\mathrm{NO_2}$ reacting on photo-sensitised organics (R11), and to a
lesser extent from $\mathrm{NO_2}$ disproportionation (R9), which would further increase $Y(\mathrm{HONO})$ and bring the HONO production rate from nitrate photolysis closer to the measured flux density. As well as the stated uncertainties in $z_{\mathrm{e}}$ and $w(\mathrm{NO_3}^-)$, there is uncertainty in the quantum yield for nitrate photolysis in snow: an error of $50\,\%$ has been calculated from Chu and Anastasio (2003). These uncertainties are represented by the red shading in Fig. 7.

Other assumptions made when carrying out this calculation include that the light attenuation in snow is exponential and that
the snow density is $0.3\,\mathrm{g\,cm^{-3}}$, with a nitrate mass fraction that does not change with depth in the snow pack. It has also been assumed that all HONO produced will be released from the snow immediately; snow-pack produced HONO can be vented via wind pumping from the open snow pore space into the air above (Liao and Tan, 2008). The largest gradient in HONO amount fraction with height observed occurs during a wind speed increase (see Fig. 6) suggesting such wind pumping does occur at Halley.

Photochemical reaction of $\mathrm{NO_2}$ on organic surfaces in the snow is a commonly suggested HONO formation mechanism (R10-R11; Ammann et al. (2005); George et al. (2005)). There is likely to be significant photosensitised organic matter present in the snow at Halley. Calace et al. (2005) found fulvic acid mass concentrations between 16 and $400\,\mu\mathrm{g\,L^{-1}}$ in coastal snow in east Antarctica, and Antony et al. (2011) found the total organic carbon (TOC) concentration of surface snow was 88 to $928\,\mu\mathrm{g\,L^{-1}}$ along a transect to the coast in east Antarctica with the higher values measured nearer the coast which they at-
tributed to marine sources associated with sea-spray. $\mathrm{NO_2}$ has been measured previously at Halley (see Table 3), amount fractions were lower than at other sites, regularly $<10\,\mathrm{pmol\,mol^{-1}}$, which would limit HONO production via this mechanism. However, in interstitial air in snow blocks at Neumayer station (a similar coastal ice shelf location) $\mathrm{NO_2}$ amount fractions were found to be higher than in ambient air, up to $40\,\mathrm{pmol\,mol^{-1}}$ (Jones et al., 2000). In their laboratory study of this HONO production mechanism, Bartels-Rausch et al. (2010) estimate that such an $\mathrm{NO_2}$ amount fraction, with a snow TOC concentration
of 10 to $1000\,\mu\mathrm{g\,L^{-1}}$, would lead to a flux density of $3 \times 10^{12}$ to $4 \times 10^{12}\,\mathrm{m^{-2}\,s^{-1}}$. This estimate agrees well with the mea-





sured HONO flux density from the snow at Halley, provided that all HONO produced is also emitted into the atmosphere. To investigate this possible source further the snow-pack must be analysed in more detail for the presence of such photosensitised species.

## 4.2 Additional HONO source

In order to assess the consistency of the measured HONO amount fractions and flux density, a simple box model calculation was undertaken. The change in the atmospheric HONO amount fraction over time can be written as the sum of the main sources and sinks:

$$\frac{d[\text{HONO}]}{dt} = k_4[\text{NO}][\text{OH}] + \frac{P_{\text{ss}}(\text{HONO})}{h} - J(\text{HONO})[\text{HONO}] - k_5[\text{OH}][\text{HONO}] \tag{7}$$

which takes into account the production of HONO from photolysis of snow nitrate ($P_{\text{ss}}(\text{HONO})$), as well as HONO formation
through NO and OH combination (R4) and loss through reaction with OH (R5). This can be simplified as the rates of reactions R4 and R5 are typically slow, especially under remote conditions, meaning that in this model the HONO budget is dominated by emission of HONO from snow nitrate photolysis and atmospheric photolysis of HONO itself. Rearranging and simplifying Eq. (7) gives

$$P_{\text{ss}}(\text{HONO}) = h \times \left( \frac{d[\text{HONO}]}{dt} + J(\text{HONO})[\text{HONO}] \right). \tag{8}$$

For simplification it is assumed that the emitted HONO is homogeneously mixed in a layer of height $h$. The boundary layer height at Halley can be hard to define (Anderson and Neff, 2008) but for the current measurement period the boundary layer height is likely above 40 m (King et al., 2006; Jones et al., 2008). These calculated flux densities, for $h$ between 10 and 50 m, reflect the shape of the measured flux density well showing peaks at noon, though the calculated flux densities appear to decrease to 0 at night which the measurements do not (Fig. 7).

The assumption of a constant HONO amount fraction up to height $h$ above the snow surface is the largest source of uncertainty in this simple box model. Steep gradients in HONO amount fraction are expected, caused by the ground surface source, the turbulent transport, and the photolytic loss of HONO in the atmosphere. The gradients were confirmed in the present study, for which the HONO amount fraction decreased to ca. half between 0.24 and 1.26 m height (see Fig. 6 (b)). However, these gradients can only be described correctly by a 1D model approach, which is out of scope of the present study. Errors in the flux
density calculation caused by deviations from MOST are also a possibility, and this comparison is further limited by the flux density measurements only being possible for one 12-hour period.

For completeness, we tried including reactions involving NO and OH (R4 and R5) in the simple box model represented by Eq. (7). Due to the lack of concurrent observations, previous measurements of other gases during the CHABLIS campaign in 2004 and 2005 at Halley (Jones et al., 2008) were used for further calculations. Specifically, OH and NO data for the days of
January and February 2005 corresponding to the days HONO was measured on in 2022 (22 January to 3 February) have been used here. NO amount fractions were low, with a mean of 5.7 pmol mol$^{-1}$, but showed a diurnal cycle peaking between 1900 and 2000 UTC, 5 hours after solar noon (1400 UTC) (Bauguitte et al., 2012). The mean OH concentration was $3.9 \times 10^5$ cm$^{-3}$





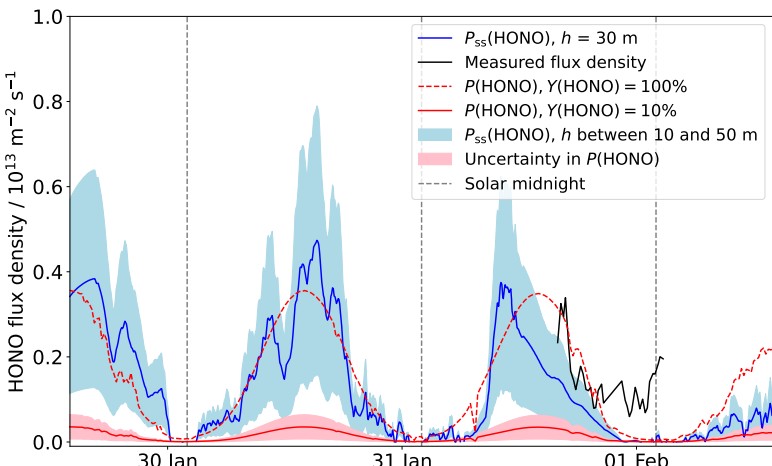

**Figure 7.** The HONO production from from Eq. (8), $P_{ss}(HONO)$, which assumes that the only HONO source is nitrate photolysis in the snow and the only sink is HONO photolysis is plotted. The blue filled region is the production expected for mixing heights of 10 m and 50 m. The HONO production from nitrate photolysis, $P(HONO)$, is plotted assuming a HONO yield of 100 % and 10 %. The uncertainty in this, from $z_e$, $w(NO_3^-)$ and the photolysis quantum yield, is represented by the red filled region. The measured flux density is also plotted. The production of HONO through reaction of $OH + NO$ (R4) is not shown as its contribution would be $< 10^{11} \, m^{-2} \, s^{-1}$.

**Table 3.** Observations of $HO_x$ concentrations and $NO_x$ amount fractions made during the CHABLIS campaign at Halley (January – February 2005) and the temperature ($\theta$) and atmospheric pressure ($p_{atm}$) observed during this campaign (January - February 2022).

| Species: | $[OH]/cm^{-3}$ | $[HO_2]/cm^{-3}$ | $y(NO)/pmol \, mol^{-1}$ | $y(NO_2)/pmol \, mol^{-1}$ | $\theta/°C$ | $p_{atm}/hPa$ |
|---|---|---|---|---|---|---|
| Mean | $3.9 \times 10^5$ | $2.0 \times 10^7$ | 5.7 | 4.1 | $-4.0$ | 986 |
| Range | $(0.8 - 7.9) \times 10^5$ | $(0.5 - 4.0) \times 10^7$ | $< 5 - 67$ | $< 5 - 70$ | $-12.9$ to $+1.1$ | 972 to 995 |
| Ref. | Bloss et al. (2007) | | Bauguitte et al. (2012) | | | |

with an average noontime maximum of $7.9 \times 10^5 \, cm^{-3}$ (Bloss et al., 2007). These data, along with other gases that were measured in the campaign are summarised in Table 3. The reaction rate coefficients used in these calculations are summarised in Table 2. Using these NO and OH concentrations, a new value of $P_{ss}(HONO)$ was calculated. As expected, the inclusion of these reactions does not make a large difference; the flux density calculated by this method is on average 3 % smaller than that from Eq. (8), though it is also occasionally larger.


### 4.3 Photo-stationary-state HONO

If the flux density from the snow is ignored, the photo-stationary-state (PSS) HONO amount fraction can be calculated. This assumes HONO is solely formed in the air through reaction R4 and lost through reactions R1 and R5.

$$\frac{d[\text{HONO}]}{dt} = 0 = k_4[\text{NO}][\text{OH}] - J(\text{HONO})[\text{HONO}] - k_5[\text{OH}][\text{HONO}] \tag{9}$$

Using NO and OH CHABLIS data again, an average photo-stationary-state HONO amount fraction of $0.07 \, \text{pmol} \, \text{mol}^{-1}$ was calculated. This showed a diurnal cycle with a maximum at solar noon and minimum at night. However, this calculation is only valid at the HONO measurement height of $0.4 \, \text{m}$ if it is assumed there are no gradients in the NO and OH amount fractions which were measured at higher altitudes, $4.5$ to $6 \, \text{m}$ above the snow. HONO was found to have a steep amount fraction gradient which suggests that by $4.5$ to $6 \, \text{m}$ above the snow the its amount fraction could be close to the photo-stationary-state amount fraction.

The inclusion of HONO formation through a dark reaction of $\text{NO}_2$ on surfaces (R10) (Ammann et al., 2005) in the PSS calculation raised the HONO amount fraction to $0.3 \, \text{pmol} \, \text{mol}^{-1}$ which is still significantly lower than that measured. For this reaction $k_{10} = 1.0 \times 10^{-5} \, \text{s}^{-1}$ was estimated from the night-time increase in the $\text{HONO} : \text{NO}_x$ ratio in the average diurnal cycle observed at Halley, see Table 2 (Kleffmann et al., 2003). Clearly an additional source is required to raise the HONO amount fraction above stationary-state levels.

### 4.4 HONO : $\text{NO}_x$ Ratio

The $\text{HONO} : \text{NO}_x$ amount fraction ratio can provide a check on the HONO data: Under steady state conditions of HONO and $\text{NO}_x$ sources and sinks, as well as assuming that all $\text{NO}_x$ is produced by HONO photolysis as an upper limit, the $\text{HONO} : \text{NO}_x$ ratio should approach that of their lifetimes ($\tau(\text{HONO}) : \tau(\text{NO}_x)$) (Villena et al., 2011).

Using the HONO data collected and the $\text{NO}_x$ amount fraction for the same time period in 2005 (Bauguitte et al., 2012), the ratio of the amount fractions was calculated. This is between $0.15$ and $0.35$ and shows no diurnal cycle. This is significantly lower than other studies (Beine et al., 2001, 2002; Dibb et al., 2002; Jones et al., 2011; Legrand et al., 2014) supporting that our measurements are comparatively free from interferences. Only during measurements in Barrow, Alaska, were even lower $\text{HONO} : \text{NO}_x$ ratios of $0.06$ observed, also using the LOPAP technique (Villena et al., 2011). However, the 2022 HONO measurements were made significantly closer to the snow surface than the 2005 $\text{NO}_x$ measurements ($0.4 \, \text{m}$ compared to $6 \, \text{m}$). The steep gradient in HONO that has been observed suggests that the HONO amount fraction at $6 \, \text{m}$ above the snow will be considerably lower. This would further reduce the ratio, which still supports that these measurements are relatively free from interferences.

The ratio of HONO to $\text{NO}_x$ lifetimes is $0.07$ at night (80 minutes:19 hours, $\tau(\text{HONO})$ calculated for loss by photolysis and $\tau(\text{NO}_x)$ for loss by reaction with OH (Seinfeld and Pandis, 1998)). This is a factor of $2 - 4$ lower than the measured $\text{HONO} : \text{NO}_x$ ratio. The daytime ratio of lifetimes (12 minutes:6 hours (Bauguitte et al., 2012)) is $0.03$ and is even lower compared to the range measured because the HONO measurements were made close to the snow surface (the HONO source) where the steady-state conditions are not fulfilled.

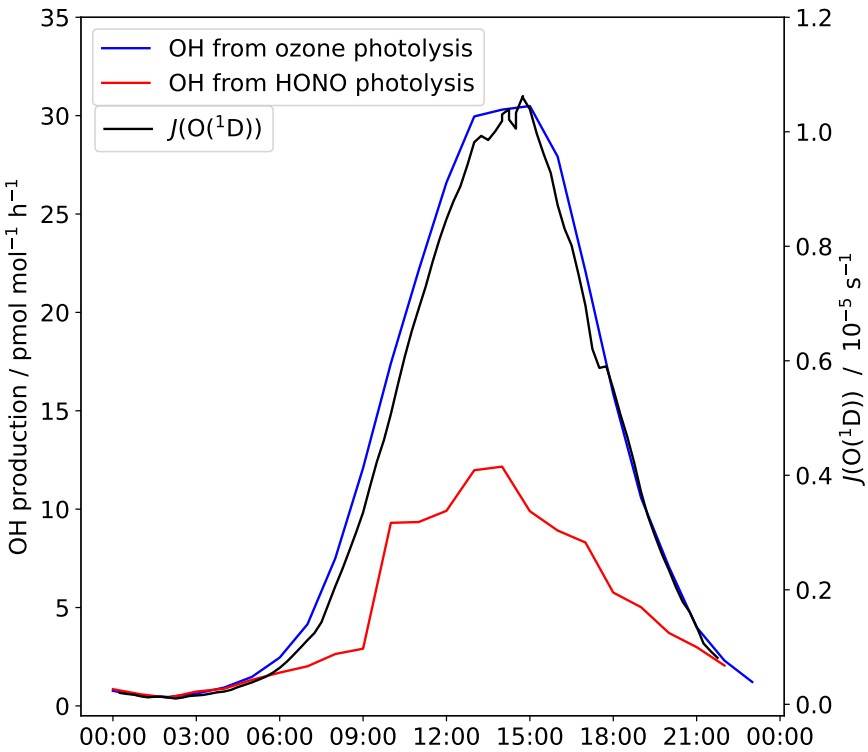

**Figure 8.** The contribution of HONO (red) and ozone (blue) to the OH production rate at Halley. The average $J(\mathrm{O}(^1\mathrm{D}))$ for Halley is plotted in black.

## 4.5 HONO as a source of OH

HONO photolysis is an OH source; when the measured HONO amount fraction is higher than the calculated photo-stationary-state amount fraction, HONO is a net source of the OH radical. The importance of this net source compared to that from ozone at Halley is plotted in Fig. 8.

As part of the CHABLIS campaign, Bloss et al. (2007) calculated the contribution of various OH sources to the $\mathrm{HO}_x$ budget at Halley. It was found that the halogen species contributed the most; Halley's coastal location means there are high halogen concentrations (Saiz-Lopez et al., 2007). The halogens species HOI and HOBr show a strong diurnal pattern and contribute up to 89 and 14 $\mathrm{pmol\,mol^{-1}\,h^{-1}}$ OH at solar noon respectively. The OH production by HOBr is comparable to that calculated for HONO from the measurements made here (12 $\mathrm{pmol\,mol^{-1}\,h^{-1}}$). However, it must again be highlighted that the halogen measurements were made higher above the snow (4 to 5 m, (Saiz-Lopez et al., 2008)) than those of HONO and that the HONO





**Table 4.** Solar noon and average OH production by HONO, ozone and halogen species at different Antarctic (Halley and Dome C) and Arctic (Barrow) locations.

| Location | OH source | Maximum OH production / pmol mol$^{-1}$ h$^{-1}$ | Average OH production / pmol mol$^{-1}$ h$^{-1}$ |
|---|---|---|---|
| Halley | HONO | 12.2 | 4.9 |
| (this campaign) | Ozone | 30.5 | 11.6 |
| | | | |
| Halley | Ozone | 62 | |
| (CHABLIS campaign | HOI | 89 | |
| (Bloss et al., 2007) | HOBr | 13.7 | |
| | | | |
| Dome C | HONO | 298 | 194 |
| (Kukui et al., 2014) | Ozone | 23 | 4.1 |
| | | | |
| Barrow | HONO | 90 | |
| (Villena et al., 2011) | Ozone | 2.7 | |

OH source estimate is only valid at 0.4 m where the HONO was measured. At higher altitudes above the snow HONO will approach the PSS amount fraction leading to a lower contribution to the OH radical source. Thus, the contribution of HONO to the OH radical initiation will be limited only to the lowest part of the mixing layer.

The surface ozone amount fraction recorded during this measurement period is comparable with that measured during the CHABLIS campaign (mean 14 nmol mol$^{-1}$ here, 10 nmol mol$^{-1}$ during January 2005 ) and lower than at other Antarctic sites due to the high halogen concentrations. However, at the other sites in Table 4, HONO is a more important OH source than ozone. The low temperatures at these sites (mean: Barrow $-26$, DC $-31\,°C$) mean the extremely dry air limits OH production via reactions R2 and R3. At Halley, a warmer coastal location, the water vapour concentration is not limiting and OH formation through ozone photolysis still dominates. Nevertheless, HONO photolysis as a source of OH cannot be discounted. The HONO amount fractions that were initially thought to be present at Halley, detected during the CHABLIS campaign (Bloss et al., 2010), would lead to an over-estimation of the HO$_x$ budget (Bloss et al., 2007). Besides conceptional problems when comparing box model calculations with HONO measurements close to the snow surface, these HONO measurements should make the HO$_x$ budget at Halley easier to rationalise.

## 5 Conclusions

We have presented the first interference-free measurements of atmospheric HONO amount fractions at an Antarctic coastal ice-shelf location. The lower values we observed here may be at least partly due to interference correction by the two-channel LOPAP technique. Amount fractions were between $< 0.26$ and $14$ pmol mol$^{-1}$, with a mean of $2.1$ pmol mol$^{-1}$, and exhibited





a diurnal pattern peaking at noon. A HONO flux density of 0.5 to $3.4 \times 10^{12}\,\mathrm{m^{-2}\,s^{-1}}$ from the snow was measured which is at the upper limit of the estimated HONO production from nitrate photolysis suggesting this reaction is a driver of HONO release

from the snow. The flux density required to reach the measured amount fraction, with known HONO sources and sinks, was calculated by a simplified box model and is comparable to that measured here. HONO is an important OH source at Halley: these measurements suggest that HONO could contribute up to $12\,\mathrm{pmol\,mol^{-1}\,h^{-1}}$ of OH which should fit better with the $HO_x$ budget previously modelled (Bloss et al., 2007, 2010). However, such calculations were limited by the strong HONO gradients and by the height difference between the HONO measurements of this campaign and the $HO_x$, $NO_x$ and halogen

species measurements of the CHABLIS campaign.

There is a clear need for a complete campaign covering $HO_x$, $NO_x$, $NO_y$ and halogen species, with measurements at the same height, to understand the interaction of the snow surface and boundary layer above. The observation of a steep gradient in HONO amount fraction requires further investigation. A 1D model combining amount fractions and fluxes of such gases, as well as meteorological data, is crucial for forming a consistent picture of the importance of the snow in the composition of the

air at different heights through the polar boundary layer.

## Appendix A: Flux Calculations

As discussed in the main text, the flux-gradient method was used to determine the HONO flux density using Eq. (1):

$$F = -K_\mathrm{c}\frac{\mathrm{d}c}{\mathrm{d}z} \tag{1}$$

where $K_\mathrm{c}$, the turbulent diffusion coefficient for a chemical tracer, may be approximated by the eddy diffusion coefficient for

heat, $K_\mathrm{h}$ (Jacobson, 2005). Using Monin-Obukhov Similarity Theory (MOST) $K_\mathrm{h}$ can be calculated via

$$K_\mathrm{h} = \frac{\kappa u_* z}{\Phi_\mathrm{h}\left(\frac{z}{L}\right)} \tag{A1}$$

where $\kappa$ is the von Karman constant (set to 0.4), $u_*$ is the friction wind velocity, $z$ is the height and $\Phi_\mathrm{h}$ the stability function for heat (Jacobson, 2005). $\Phi_\mathrm{h}$ is empirically determined as a function of $\frac{z}{L}$ where $L$ is the Obukhov length (King and Anderson, 1994).

$$L = \frac{u_*^2 \bar{\theta}}{\kappa g \theta_*} \tag{A2}$$

where $\bar{\theta}$ is the temperature, $\theta_*$ is the potential temperature scale, and $g$ is the gravitational constant. Combining Eq. (1) and (A1) results in:

$$F = -K_\mathrm{c}\frac{\mathrm{d}c}{\mathrm{d}z} = -\frac{\kappa z u_*}{\Phi_\mathrm{h}\left(\frac{z}{L}\right)}\frac{\mathrm{d}c}{\mathrm{d}z} \tag{A3}$$

which can be integrated to

$$F = -\frac{\int_{c_1}^{c_2}\kappa u_*\mathrm{d}c}{\int_{z_1}^{z_2}\Phi_\mathrm{h}\left(\frac{z}{L}\right)\frac{\mathrm{d}z}{z}} = -\frac{\kappa u_*[c(z_2)-c(z_1)]}{\int_{z_1}^{z_2}\Phi_\mathrm{h}\left(\frac{z}{L}\right)\frac{\mathrm{d}z}{z}} = \frac{\kappa u_*[c(z_1)-c(z_2)]}{\int_{z_1}^{z_2}\Phi_\mathrm{h}\left(\frac{z}{L}\right)\frac{\mathrm{d}z}{z}}. \tag{A4}$$





This is the same as Eq. (2) in the main text. Therefore, to find the flux density the amount fraction of the gas must be known at two heights, along with the integrated stability function and $u_*$. 3D sonic anemometer measurements are normally used to find $u_*$ but were not available for this measurement period so $u_*$ was first estimated for a neutral boundary layer according to (Anderson and Bauguitte, 2007)


$$u_* = \frac{\kappa u(z_r)}{\ln \frac{z_r}{z_0}} \tag{A5}$$

where $u(z_r)$ is the wind speed measured at height $z_r$, and $z_0$ is the surface roughness length that has been measured previously at Halley, $z_0 = (5.6 \pm 0.5) \times 10^{-5}$ m (King and Anderson, 1994). Forms of the integrated stability function have been established for stable and neutral conditions above snow. The value of $\frac{z}{L}$ provides an indication of the boundary layer conditions and hence the expression for the integrated stability function. $L$ was estimated by Eq. (A2) which requires $\theta_*$. This was also initially

estimated for a neutral boundary layer, using temperature measurements made at two heights (Jacobson, 2005):

$$\theta_* = \frac{\kappa [\bar{\theta}_2 - \bar{\theta}_1]}{\int_{z_{\theta_1}}^{z_{\theta_2}} \Phi_h(\frac{z}{L})\frac{dz}{z}}. \tag{A6}$$

In the case of the measurement period at Halley the value of $\frac{z}{L}$ was found to be close to zero so the boundary layer was assumed to be neutral and the initial estimates of $u_*$ and $\theta_*$ were valid. The integrated stability function for a neutral boundary layer is:

$$\int_{z_1}^{z_2} \Phi_h(\frac{z}{L})\frac{dz}{z} = Pr_t \ln \frac{z_2}{z_1} \tag{A7}$$

where $Pr_t$ is the turbulent Prandtl number with a value of 0.95 (King and Anderson, 1994). Combining this with Eq. (A4) gives

$$F = \frac{\kappa u_* [c(z_1) - c(z_2)]}{Pr_t \ln \frac{z_2}{z_1}} \tag{A8}$$

which was used to calculate the flux density.

*Author contributions.* Field measurements at Halley were carried out by AMHB with assistance from FAS. Data analysis was done by
AMHB with supervision from MMF, Ja K, Jö K and AEJ. AMHB wrote the manuscript first draft with contributions from all co-authors.

*Competing interests.* One of the (co-)authors is a member of the editorial board of Atmospheric Chemistry and Physics.

*Acknowledgements.* This work was supported by the Natural Environment Research Council and the ARIES Doctoral Training Partnership
[grant number NE/S007334/1]. We acknowledge the Collaborative Antarctic Science Scheme (CASS) for funding the fieldwork at Halley VI
Research Station. We thank the British Antarctic Survey Halley science team for their support provided during the field season, in particular



Thomas Barningham and Jack Farr. We also thank BAS engineers Ross Sanders and Rad Sharma for designing and building the LOPAP

elevator.





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
