# Peer review of "Snowpack nitrate photolysis drives the summertime atmospheric nitrous acid (HONO) budget in coastal Antarctica"

_Atmospheric Chemistry and Physics, 2022_

## Referee Comment (RC1)

Review of « Snowpack nitrate photolysis drives the summertime atmospheric nitrous acid (HONO) budget in coastal Antarctica » by Amelia M. H. Bond, Markus M. Frey, Jan Kaiser, Jörg Kleffmann, Anna E. Jones, and Freya A. Squires

The manuscript of Bond et al. presents atmospheric and flux density measurements of nitrous acid (HONO) made for the first time at the coastal Antarctic Halley station over 10 days during the austral summer 2021/22. Previous studies which dealt with HONO measurements in Antarctica, revealed significant measurement problems, resulting in overestimations of HONO. The measurement technique used in the present study is a two-channel Long Path Absorption Photometer (LOPAP) which limits numerous interferences except for HNO4, a species that is mostly present in low temperature atmospheres.
Atmospheric HONO measurements at Halley were made over 10 days including around 12hours during which the measurement height was changed between two heights to estimate the HONO flux out of snow. After presenting the HONO production and destruction mechanisms, the site and the used methods are presented.
Then the newly gained HONO results are discussed against already existing HONO measurements in Antarctica and in view of what would have expected from the known production and destruction mechanisms.

This study fills an important information gap about HONO sources and sinks in the polar boundary layer at a snow-covered coastal site, a key question for the oxidative properties of the polar atmosphere. Thus, the manuscript is clearly in the scope of ACP. The study presents new data and the manuscript is structured adequately with respect to the aim of the manuscript. In my opinion the manuscript is suitable for publication after one point has been corrected.

The only real concern I have is that the authors invoke in section 4.1 (paragraph which starts at line 295) very high levels of organic matter at Halley by referencing Calace et al. (2005) and Antony et al. (2011) to explain the flux density measured during the field campaign, whereas Legrand et al. (2013) clearly demonstrated that these studies overestimate the organic matter content significantly. Instead, Legrand et al. (2013) reports much lower levels of about 10-20 ppbC of dissolved organic carbon at inner continental sites as well as near coastal sites (see also Figure 3 for HULIS species). Thus, the contribution of the production mechanism via R10 and R11 are likely too limited to explain the observed HONO flux.
On the other hand, concerning the organic matter content which should be low at both sites, snow at Halley and Dome C might not be so different, what might allow to do a first order estimation of the Halley HONO flux density via the Halley NOx flux measured during the CHABLIS campaign and the HONO to NOx production rate ratio measured in the Dome C snow photolysis experiment described by Legrand et al., 2014.
Such an exercise could give a hint whether the, to a few hours limited, HONO flux measurements conducted within this study would be representative or not.

Minor comment:

1) The method sections 2.2. and 2.4 are rather long considering that both methods (LOPAP and flux calculations) are already reported in literature. For the shake of the straightness of the manuscript, the authors might consider to shorten these topics considerably in the main manuscript and to detail them in a supporting text. On the other hand, the manuscript is not too long, as it is, therefore I leave the decision to the authors.

2) Figure 5 (line 201) is addressed for the first time before Figure 4 (line 205) is addressed for the first time. Therefore figure 4 and 5 might be inversed in their order.

3) Please let the reader know where the data of this study will be available

References:
Legrand, M., Preunkert, S., Jourdain, B., Guilhermet, J., Fain, X., Alekhina, I., Petit, J.R., (2013). Water-soluble organic carbon in snow and ice deposited at Alpine, Greenland, and Antarctic sites: a critical review of available data and their atmospheric relevance. Climate of the Past. 9. 10.5194/cp-9-2195-2013.

---

## Author Comment (AC1)

We thank the referees for their time reading the manuscript, the supportive comments and suggestions that have improved the clarity of the manuscript.

Please find below a detailed response to each suggestion. The reviewers' comments and our responses are labelled. Text from the manuscript is in quotations and with changes highlighted in blue.

**REVIEWER 1**

**Comment:**

This study fills an important information gap about HONO sources and sinks in the polar boundary layer at a snow-covered coastal site, a key question for the oxidative properties of the polar atmosphere. Thus, the manuscript is clearly in the scope of ACP. The study presents new data and the manuscript is structured adequately with respect to the aim of the manuscript. In my opinion the manuscript is suitable for publication after one point has been corrected.

The only real concern I have is that the authors invoke in section 4.1 (paragraph which starts at line 295) very high levels of organic matter at Halley by referencing Calace et al. (2005) and Antony et al. (2011) to explain the flux density measured during the field campaign, whereas Legrand et al. (2013) clearly demonstrated that these studies overestimate the organic matter content significantly. Instead, Legrand et al. (2013) reports much lower levels of about 10-20 ppbC of dissolved organic carbon at inner continental sites as well as near coastal sites (see also Figure 3 for HULIS species). Thus, the contribution of the production mechanism via R10 and R11 are likely too limited to explain the observed HONO flux.

On the other hand, concerning the organic matter content which should be low at both sites, snow at Halley and Dome C might not be so different, what might allow to do a first order estimation of the Halley HONO flux density via the Halley $NO_x$ flux measured during the CHABLIS campaign and the HONO to $NO_x$ production rate ratio measured in the Dome C snow photolysis experiment described by Legrand et al., 2014. Such an exercise could give a hint whether the, to a few hours limited, HONO flux measurements conducted within this study would be representative or not.

**Response:**

We added a discussion of the findings of Legrand et al. (2013) in the text, including that the organic content of the snow could not be as high as initially suggested. We have done the calculation suggested and included it in the text:

'…marine sources associated with sea-spray. Legrand et al. (2013) have highlighted that these studies could overestimate the organic matter content due to their sampling method and measurement technique. They suggest that the organic matter at coastal Antarctic sites could be lower, comparable to inland sites like Dome C ($3 - 8$ µg $L^{-1}$). Legrand et al. (2014) suggest that this could still lead to significant HONO production. Assuming Dome C and Halley snow have similar organic content, a HONO flux density can be estimated based on the HONO:$NO_x$ emission ratio measured in a laboratory study of Dome C snow (Legrand et al., 2014) and the measured $NO_x$ flux density at Halley (Bauguitte et al., 2012). The HONO:$NO_x$ ratio is temperature dependent; the highest temperature studied by Legrand et al. (2014) is $-13°C$ which is below the Halley air temperature for the flux measurement period. An emission ratio

of 0.77 and $NO_x$ flux density of $7.3 \times 10^{12}$ m$^{-2}$ s$^{-1}$ give a HONO flux density of $5.6 \times 10^{12}$ m$^{-2}$ s$^{-1}$, close to the measured value. $NO_2$ has been measured…'

**Comment:** The method sections 2.2. and 2.4 are rather long considering that both methods (LOPAP and flux calculations) are already reported in literature. For the shake of the straightness of the manuscript, the authors might consider to shorten these topics considerably in the main manuscript and to detail them in a supporting text. On the other hand, the manuscript is not too long, as it is, therefore I leave the decision to the authors.

**Response:** We have decided to leave the method section as is. ACP's manuscript guidelines require methodological details to be part of the main text; they should not appear as supplement.

**Comment**: Figure 5 (line 201) is addressed for the first time before Figure 4 (line 205) is addressed for the first time. Therefore figure 4 and 5 might be inversed in their order.

**Response:** Figures 4 and 5 have now been reversed in order.

**Comment:** Please let the reader know where the data of this study will be available.

**Response:** The data will be available at the UK Polar Data Centre. This is now stated in the manuscript.

**REVIEWER 2**

**Comment:**
This paper reports the results of a field measurement study assessing the sources and contribution of HONO to oxidation capacity of the Antarctic boundary layer at a coastal (ie, sea level altitude) location. This remains a persistent challenge and key issue in understanding within- and above-snowpack atmospheric chemical processing. The paper combines field measurement results from a challenging environment with derivation of vertical flux using the gradient flux approach, and simple calculations to assess the contribution of HONO to OH formation.

The results are similar to those reported from other comparable locations, and advance quantitative understanding of the importance of HONO at this location; they go some way to unpicking conflicting results (from HONO measurements likely over-estimated previously) at this location.

The measurements appear to have been carefully performed with appropriate corrections and blanks etc, and are described at an appropriate level of detail. The analysis presented is carefully considered. Use of species measurements from previous campaigns (i.e. different

years) is made, but this is unavoidable and appropriately noted / caveated. The paper is well written and clearly presented.

I recommend the paper is accepted for publication, subject to the authors considering the corrections / suggestions outlined below.

A style point: personally I find the terminology "amount fraction" somewhat jarring – "mixing ratio" preferable. We need to make our manuscripts accessible as well as precise ! But this is up to the Editor and journal…

**Response:** ACP expects authors to follow IUPAC terminology. "Amount fraction" (short for "amount-of-substance fraction") are the terms recommended in the IUPAC Green Book. Mixing ratio is an ambiguous term and therefore best avoided (e.g., it can refer to mass, volume and amount fractions, or even mass or amount per volume of air). We have therefore kept the term "amount fraction".

**Comment:** R4 is termolecular and should include the third body M – line 31 and subsequently in the manuscript.

**Response:** R4 has been amended to include M.

**Comment:** L51 are reactions "accelerated" by sunlight – consider phrasing.

**Response:** This has been rephrased as follows:
'The uptake of $NO_2$ on such organics is greater in the presence of sunlight (George et al., 2005)'

**Comment:** L81 / L232 it would be possible to estimate the magnitude of potential PNA interference – using the previous data for $HO_2$ and $NO_2$ to estimate $[HO_2NO_2]_{ss}$ and hence the interference contribution. This might usefully be added to the discussion.

**Response:** The average steady state $HNO_4$ amount fraction was calculated as 0.05 pmol mol$^{-1}$. Legrand et al. (2014) suggest 100 pmol mol$^{-1}$ of $HNO_4$ would lead to an overestimate by 15 pmol mol$^{-1}$ HONO. The interference is therefore insignificant and below the detection limit of the LOPAP. This will be included in the manuscript:
'The LOPAP's response to $HNO_4$ has been investigated in both the laboratory with an $HNO_4$ source and in the field at Dome C by placing a heated tube at the instrument inlet to decompose $HNO_4$. Both showed that the LOPAP partially measures $HNO_4$ as HONO with approximately 100 pmol mol$^{-1}$ $HNO_4$ leading to an interference of 15 pmol mol$^{-1}$, but further investigation is needed to systematically quantify this effect (Legrand et al., 2014).'
'As a further check on this interference, the steady-state concentration of $HNO_4$ was calculated. The method for this is detailed in Appendix B. Again the concentrations of $HO_2$, $NO_2$ and OH from the CHABLIS campaign were used. The average steady-state amount

fraction was 0.05 pmol mol$^{-1}$. Using the estimate of Legrand et al. (2014), this suggests that the interference is likely <0.01 pmol mol$^{-1}$, well below the detection limit of the LOPAP.'

**Comment:** L113 the key assumption of LOPAP is that HONO is effectively entirely removed in coil 1, but that the abundance of interferents is effectively unchanged – so that subtraction of coil 2 signal from coil 1 signal results in just HONO.

**Response:** The fact that this is an assumption has been emphasised in the text:
'The interferences  are assumed to be taken up to the same small extent in both channels so that the HONO amount fraction can be calculated by subtracting the signal in channel 2 from that in channel 1'

**Comment:** L139 How did the 15 min height change compare with the (liquid) residence time of the LOPAP – were the data used for flux calculations adjusted / truncated for the delay from instrument residence time between gas intake and absorption signal response.

**Response:** The average response time of the LOPAP (90 % of final signal) was (8.0 ± 1.5) min. All LOPAP data was shifted to account for the time delay, including the flux data. This is now stated in the text:
'…at the instrument inlet. The detection limit ($3\sigma_{blank}$) was 0.26 pmol mol$^{-1}$ for the measurement period. The average response time (90 % of final signal change) was (8.0 ± 1.5) min.'
'The elevator is depicted in Fig. 3. The LOPAP data were shifted to account for the time delay ((17 ± 2) min) between gas intake and the observed absorption signal. This is determined from the average of all abrupt concentration changes (start/stop of blanks) and defined as the time between concentration change and the 50 % response of the instrument.'

**Comment:** L198 what albedo assumed for the TUV calculations.

**Response:** 0.95, this has been included in the text:

'*F* is the actinic flux derived from the TUV radiation model over the wavelength range 300 to 1200 nm using measured ozone column density, a surface albedo of 0.95, and assuming clear sky conditions (Madronich and Flocke, 1999; Lee-Taylor and Madronich, 2002).

**Comment:** L291 worth commenting on the snow surface age vs porosity (ie fresh snow or subject to many weeks freezing or…)

**Response:** This is now mentioned in the text as a factor to consider when evaluating wind pumping:

'… suggesting such wind pumping does occur at Halley. The degree of wind pumping will be affected by snow permeability, which is related to snow porosity (Waddington et al., 1996). During this measurement period the snow was fresh and therefore more porous and likely more permeable than aged snow.'

**Comment:** Fig 7 – caption - not sure that $P_{ss}$(HONO) makes any assumption about the nature of the source.

**Response:** This is true, the figure caption has been amended:

'HONO production calculated from Eq. (8), $P_{ss}$(HONO).'

**Comment:** Table 3 – what is y ?

**Response:** Amount fraction, this has been included in the table caption:

'Observations of $HO_x$ concentrations and $NO_x$ amount fractions (*y*) made during the CHABLIS campaign at Halley'

**Comment:** L371 the comparison of HONO and $NO_x$ lifetimes is useful – previous Halley work has shown the $NO_x$ lifetime is significantly reduced form halogen nitrate photolysis (Bauguitte et al. 2011), this will be quite different from that estimated due to $NO_2$ + OH alone – and may improve agreement with the observed HONO:$NO_x$ ratio?

**Response:** This reduction in the $NO_x$ lifetime does bring the HONO:$NO_x$ lifetime ratio closer to the measured ratio. This is now discussed in the text:

'… the steady-state conditions are not fulfilled. However, Bauguitte et al. (2012) found that the $NO_x$ lifetime was reduced by halogen processing ($BrNO_3$ and $INO_3$ formation and heterogeneous uptake). A reduced $NO_x$ lifetime would improve the agreement with the observed HONO:$NO_x$ ratio.'

**Comment:** L385 – HOI and HOBr are not primary sources of OH – they reflect $HO_x$ cycling, as they form from $HO_2$+XO, so in terms of OH sources they are really similar to $HO_2$ + NO. Suggest the table compares either primary OH sources (HONO, $O^1D$) or all OH sources (including $HO_2$ + NO etc) or (best) all $HO_x$ sources – HONO, $O^1D$, HCHO, but excluding recycling such as $HO_2$+NO, $HO_2$+XO.

**Response:** The table has been updated to include both primary OH sources and $HO_x$ recycling sources with the distinction between the two made clear in the table and caption:

**Table 4.** Maximum and average $HO_x$ production by primary $HO_x$ sources (HONO, $O_3$, formaldehyde (HCHO) and hydrogen peroxide ($H_2O_2$)) and $HO_x$ recycling at different Antarctic (Halley and Dome C) and Arctic (Barrow) locations.

| Location | OH source | Maximum OH production / $\mathrm{pmol\,mol^{-1}\,h^{-1}}$ | Average OH production / $\mathrm{pmol\,mol^{-1}\,h^{-1}}$ |
|---|---|---|---|
| Halley | $O_3$ | 31 | 12 |
| (this campaign) | HONO | 12 | 5 |
| | | | |
| Halley | $O_3$ | 62 | |
| (CHABLIS campaign, | HCHO | 10 | |
| Bloss et al. (2007)) | $H_2O_2$ | <10 | |
| | | | |
| | HOI | 89 | |
| | HOBr | 14 | |
| | $HO_2 + NO$ | 13 | |
| | | | |
| Dome C | HONO | 298 | 194 |
| (Kukui et al., 2014) | HCHO | 50 | 28 |
| | $H_2O_2$ | 28 | 12 |
| | $O_3$ | 23 | 4 |
| | | | |
| | $HO_2 + NO$ | 157 | 112 |
| | | | |
| Barrow | HONO | 90 | |
| (Villena et al., 2011) | $O_3$ | 3 | |